# Protein Hydrolysates Supplement in the Nutrient Solution of Soilless Grown Fresh Peppermint and Spearmint as a Tool for Improving Product Quality

**Danai-Christina Aktsoglou [1], Dimitrios S. Kasampalis [1], Eirini Sarrou [2], Pavlos Tsouvaltzis [1,*], Paschalina Chatzopoulou [2], Stefan Martens [3] and Anastasios S. Siomos [1]**

[1] Department of Horticulture, Aristotle University of Thessaloniki, 54124 Thessaloniki, Greece; daktsogl@agro.auth.gr (D.-C.A.); kasampal@agro.auth.gr (D.S.K.); siomos@agro.auth.gr (A.S.S.)

[2] Hellenic Agricultural Organization DEMETER, Institute of Plant Breeding and Genetic Resources, Department of Medicinal and Aromatic Plants, Thermi, 57001 Thessaloniki, Greece; esarroy@gmail.com (E.S.); chatzopoulou@ipgrb.gr (P.C.)

[3] Fondazione Edmund Mach, Research and Innovation Centre, Via E. Mach 1, 38098 San Michele all'Adige (TN), Italy; stefan.martens@fmach.it

\* Correspondence: ptsouv@agro.auth.gr; Tel.: +30-2310-998-802

**Abstract:** The present study investigated the potential of fresh peppermint (*Mentha × piperita* L.) and spearmint (*Mentha spicata* L.) production on a floating raft system combined with a commercial protein hydrolysate supplement (Amino16®) in a nutrient solution aiming to improve plant product quality. Three levels of the protein hydrolysate solution (0, 0.25 and 0.50%) were added in the nutrient solution, and the plants were harvested after twenty-four days. Plant growth characteristics were recorded, and nutritional, essential oil and polyphenolic composition were determined in fresh tissue. The addition of protein hydrolysates did not affect the fresh or dry weight but reduced plant height. Nitrate content significantly decreased, while total chlorophyll and essential oil content increased in both species. Moreover, the protein hydrolysate solution further increased total antioxidant capacity, total soluble phenol and carotenoid contents in spearmint plants, while it did not affect the essential oil and polyphenolic composition in both species. In conclusion, protein hydrolysates solution may be added in the nutrient solution, to improve the quality of peppermint and spearmint grown in a floating system, without adverse effects on crop yield or the essential oil and polyphenolic profile.

**Keywords:** amino acid solution; bio-stimulant; hydroponics; secondary metabolite

## 1. Introduction

Peppermint (*Mentha × piperita* L.) and spearmint (*Mentha spicata* L.) are aromatic perennial herbs that belong to the Lamiaceae family, and both contain essential oil with high economic value [1]. Peppermint is a natural sterile hybrid between *M. aquatica* L. and *M. spicata* L. native in India, while spearmint is believed to have originated in Europe [2,3]. However, both are cultivated in many countries, namely in the USA, India and China [1]. The chemical composition of spearmint essential oil may vary widely among the natural populations, resulting in nine chemotypes, depending on the main essential oil components, with carvone and limonene type being the dominant ones [3]. Both plants produce stolons that grow aboveground or underground and are grown for fresh or dry use during cooking or as herbal tea, or even for their essential oils, used in the food, pharmaceutical and perfumery industries [1–3]. They are also rich in phytonutrients, such as vitamins, minerals and natural antioxidants, which are beneficial to health, as they possess nutritional, antioxidant and pharmaceutical properties [3,4]. The main antioxidant components of peppermint and spearmint are the essential oil, polyphenols, vitamins and carotenoids [2,4].

Consumer needs for convenient products are increasing, due to their limited available time for cooking, along with their need for healthier and safer food consumption, with the lowest environmental impact [5,6]. In this context, the fresh-cut industry provides new products to meet the consumer needs for fresh, convenient, safe products of high quality that retain their aroma and flavor [7]. Fresh-cut herbs are an alternative marketing strategy, as they retain more of their aroma than dry products, thus adding value to the packaged product [8,9].

In order to maximize production of high-quality herbs and reduce inputs while adopting sustainable agricultural practices, more and more growers favor a soilless production system [10] and the use of bio-stimulants [11]. Such a production system is the floating system, which has many advantages, such as simplicity, functionality, low operational cost, high plant density, early production, optimal use of water and nutrients and standardization of the production on an industrial scale [12–14]. One other advantage of the floating system is that the nutrient solution may be modified to meet the plants' needs and improve product quality [15]. Spearmint production on the floating raft system is feasible, as shown by Chrysargyris et al. [16]. Given that the requirements of peppermint and spearmint in water supply are very high [3], both species are suitable for hydroponic cultivation and, in particular, on the floating raft system.

Bio-stimulants comprise a wide range of compounds and microorganisms, such as humic acids, protein hydrolysates and nitrogen containing compounds, seaweed extracts, chitosan, inorganic compounds, mycorrhizal fungi and rhizobacteria, to name a few [11,17]. Protein hydrolysates (PHs) are amino acid mixtures produced by hydrolysis of animal and plant protein by-products [18], and there are already several studies that have confirmed the beneficial use of the PHs in many plant species. Apart from their bio-stimulant activity, the amino acids included in the PH solutions are an important alternative nitrogen source for plants [19] that may even be equally effective fertilizers as inorganic nitrogen [20]. PH solutions are mostly used on high-value crops, either added on soil or sprayed through foliar application, in order to allow plants to overcome stress conditions and promote plant growth, along with inducing the primary and secondary metabolism [18,21], but there is a limited bibliography on the inclusion of bio-stimulants in the nutrient hydroponic solution [22].

Moreover, there are only few studies that, apart from the essential oil composition and the plant growth of peppermint and spearmint grown in soilless systems, examine the nutritional quality and polyphenolic composition, as well. Therefore, the aim of this study was to investigate the effect of a commercial PH solution (Amino16®) addition in the nutrient solution of a floating raft system of both productions for fresh consumption, and specifically on the plant growth and the nutritional, the essential oil and the polyphenolic composition.

## 2. Materials and Methods

### 2.1. Plant Growth

Rooted grafts of peppermint and spearmint plants were obtained from two local populations preserved in the Department of Medicinal and Aromatic Plants of the Institute of Plant Breeding and Genetic Resources (Thermi, Greece). The grafts were transplanted at a density of 230 plants/m$^2$, on compressed polystyrene trays filled with a commercial enriched peat substrate. The trays were watered daily, for three weeks, until successful rooting, and then were trimmed, to obtain a uniform height of 5 cm in each plant.

A floating system was established in a glass non-heated greenhouse, at the farm of Aristotle University of Thessaloniki (Thermi, Greece). Temperature and relative air humidity values in the interior environment of the greenhouse, where the cultivation was held, were recorded from the beginning until the end of the experiment, with an appropriate digital recorder, HOBO U30 ONSET. Recordings were taken every minute. The mean air temperature inside the greenhouse during the cultivation ranged from 19.4 to 29.9 °C, while the mean relative humidity ranged from 49.3 to 71.4%. Nine plastic

basins (L × W × H 70 × 65 × 20 cm) were filled with 50 L nutrient solution, each with the following composition: $NO_3$-N 210 mg/L, $NH_4$-N 14 mg/L, K 391 mg/L, P 62 mg/L, Mg 49 mg/L, Ca 385 mg/L and S 232 mg/L and Fe 1150 μg/L, Mn 399 μg/L, Zn 150 μg/L, Cu 150 μg/L, B 500 μg/L and Mo 48 μg/L. The nutrient solutions (NSs) were supplemented with 0, 0.25 or 0.50% of a commercial PH solution (Amino16®, EVYP LLP, Greece). Considering that the purchase cost of such products is seriously taken into account by the growers, the dosages of the PH solution were selected to be as low as possible, in order not to significantly aggravate the production cost. Three basins were used for each PH concentration, and all nine basins were placed at a complete randomized design in the greenhouse. Two trays containing 36 peppermint and spearmint plants each were left to float on each NS. The nutrient solutions were stirred frequently, to ensure adequate aeration of the roots. Afterwards, the pH and the electrical conductivity (EC) of the NSs were measured, using a portable instrument (C5020, Consort, Turnhout, Belgium).

Twenty plants from each treatment were harvested after twenty-four days, using scissors, while still being at the vegetative herbaceous stage. Plant height and root length were measured in planta, before harvest, while plant weight was recorded after harvest. Fifty and ten grams of fresh tissue from each treatment were used for the essential oil extraction and the polyphenolic analysis, respectively. The remaining tissue was kept at −20 °C, for nutritional composition analysis.

### 2.2. Nutritional Composition Analysis

Frozen tissue of peppermint and spearmint was homogenized in a waring blender and was used to determine the nutritional composition.

Dry matter was determined after drying 30 g of homogenized tissue at 70 °C for 72 h. Total soluble solids were determined with the use of a portable electronic refractometer, (PAC-1, Atago Co Ltd., Tokyo, Japan) in juice obtained after squeezing the homogenized sample.

Nitrate content was determined by following the method by Cataldo et al. [23], in fresh tissue, using a Spectronic 20D spectrophotometer (Milton Roy Company, Ivyland, PA, USA) and a potassium nitrate standard curve.

Total antioxidant capacity was determined according to the method of Brand-Williams et al. [24], using an ascorbic acid standard curve and expressed as mg ascorbic acid equivalents (AEAC) 100 $g^{-1}$ fresh weight (f.w.). Total soluble phenols content was determined by following the method by Scalbert et al. [25], using a gallic acid curve, and expressed as mg gallic acid equivalents (GAE) $kg^{-1}$ f.w., on a Thermo Spectronic Helios Alpha UV–Vis spectrophotometer (Thermo Fisher Scientific, Waltham, MA, USA).

For the extraction of pigments, 10 mL of acetone was added to 0.2 g of homogenized tissue and incubated at −20 °C, overnight. The following day, total chlorophyll and total carotenoids content were determined, following the method by Lichtenthaler and Wellburn [26], on a UV–Vis spectrophotometer (Thermo Fisher Scientific).

### 2.3. Essential Oil

The essential oil content was determined by hydrodistillation of 50 g of fresh tissue chopped into small pieces, using a European Pharmacopoeia apparatus (Clevenger-type) for 2.5 h, with a distillation rate of 3–3.5 mL/min. The essential oil was dried over anhydrous sodium sulfate and was stored at 4–6 °C until further analysis.

Analysis was performed by Gas Chromatography–Mass Spectroscopy (GC–MS), on a Shimadzu GC 17A Ver. 3, coupled with Mass Spectrometer QP-5050A, supported by Class 5000 software. The analysis was performed on a fused silica DB-5 capillary column, with the following conditions: injection point temperature of 260 °C, ion source temperature of 200 °C and GC–MS connection temperature of 300 °C; Electron Ionization at 70 eV, scanning range 41–450 amu and scanning time of 0.50 s. The following oven temperature programs applied were: (a) 55–120 °C (3 °C/min), 120–200 °C (4 °C/min), 200–220 °C

(6 °C/min), and 220 °C for 5 min, and (b) 60–240 °C (3 °C/min). The carrier gas was He, at 54.8 kPa, and the sample inlet ratio was 1:30.

The relative to n-alkanes (C8–C20) retention indices (RIs) of the compounds were used for their identification, comparing to respective reference substances [27] and comparing the spectra to similar mass spectra of the NIST 98 and Wiley 1995 mass spectral libraries. The relative content of each compound was calculated as a percentage of the total chromatographic area.

### 2.4. Polyphenolic Compounds

Ten grams of the fresh plant tissue of each treatment was freeze-dried (Freeze-Dryer Alpha 1–2 LD plus, Christ, Osterode, Germany) at −24 °C and then pulverized to fine powder. Then 200 mg of freeze-dried tissue powder was extracted with 10 mL 80% methanol into 15 mL falcon tubes. The samples were mixed on an orbital shaker, for 3 h, at room temperature, and stored overnight, at 4 °C, in the dark. The extracts were filtered on a MILLEX 0.22 μm PTFE (Polytetrafluorethylen)membrane (Merck Millipore, Darmstadt, Germany), into a glass vial, and were analyzed for polyphenolics composition in an Ultra-Performance Liquid Chromatography coupled with a mass spectrometer (UPLC–MS/MS), as described below.

Targeted UPLC–MS/MS analysis was performed on a Waters Acquity system (Milford, MA, USA), consisting of a binary pump, an online vacuum degasser, an autosampler and a column compartment. Separation of the phenolic compounds was achieved on a Waters Acquity HSS T3 column (1.8 μm, 100 mm × 2.1 mm), kept at 40 °C. The analysis of the phenolics compounds was performed, using the method described previously by Vrhovsek et al. [28]. Mass spectrometry detection was performed on a Waters Xevo TQMS instrument equipped with an electrospray (ESI) source. Data processing was performed, using the Mass Lynx Target Lynx Application Manager (Waters).

### 2.5. Experimental Design and Statistical Analysis

The experimental design used was the complete randomized design (CRD), with three replicates per treatment, to assess the effect of the PHs concentration. One-way analysis of variance (ANOVA) was performed with SPSS v.24 (IBM) and Microsoft Excel. Significant differences among means were detected, using the least significant differences test, at $p < 0.05$. Assumptions of normality were tested, using the Shapiro–Wilk's test.

## 3. Results and Discussion

### 3.1. EC and pH of the Nutrient Solution

During the cultivation period, significant changes were observed both in EC and pH of the NSs (Figure 1). The electrical conductivity of the NSs at the time of preparation was 3.5 dS/m, and upon the addition of the low (0.25%) or the high (0.50%) PH solutions, the rate was raised to 4.0 and 4.6 dS/m, respectively. While over the majority of the growing period, the EC remained constant, twenty days beyond the start date, a rapid increase in the EC of the control NS was observed, reaching 6.2 dS/m, while a slighter increase was found in the 0.25% PHs NS solution (5.1 dS/m), and no change was exhibited in the high PHs solution, remaining constant at 4.8 dS/m (Figure 1). On the contrary, the pH of the control NS at the time of preparation was 7.2, and the addition of the PHs solution resulted in a significant decrease to 5.5 and 4.1 in the low and high dosage, respectively. The pH of the control remained rather constant during the whole cultivation period, while a slight and a rapid increase of pH started occurring within the first weeks of the growing period, in the low and high PHs solution rates, respectively, remaining constant until the end of the cultivation, at 7.5–7.7 (Figure 1).

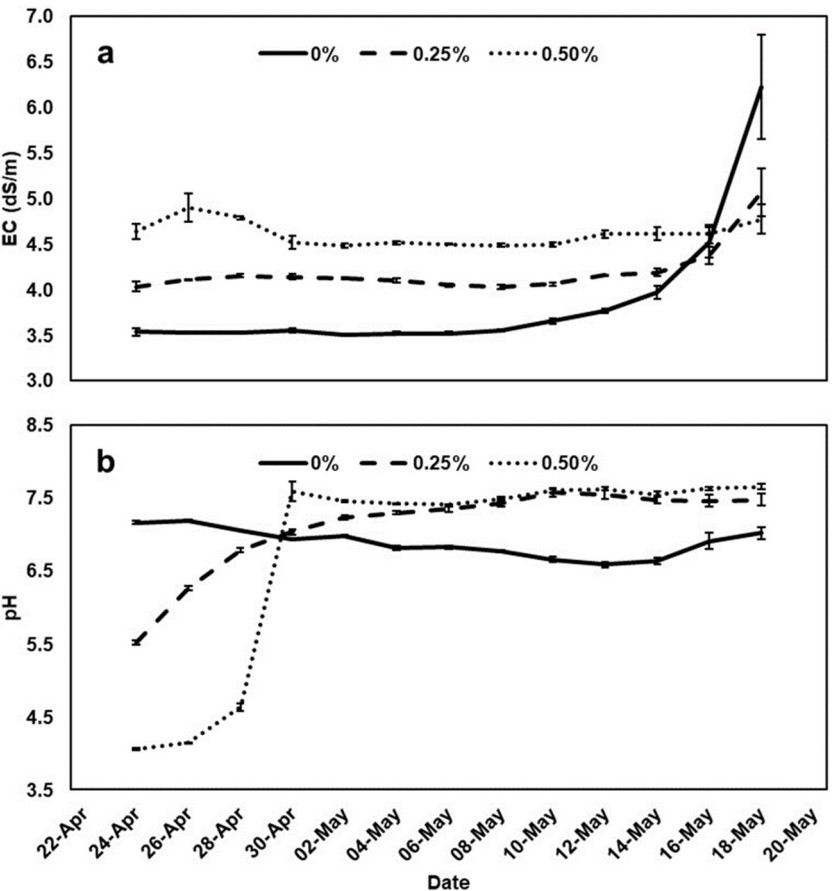

**Figure 1.** Electrical conductivity (EC) (**a**) and pH (**b**) of the nutrient solutions (NSs), in which 0, 0.25 or 0.50% of a commercial protein hydrolysate (PH) solution (Amino16®) was added during peppermint and spearmint cultivation in a floating system. Each value is a mean of three replicates.

The EC and pH of the nutrient solutions are subjected to changes during the cultivation induced by the plant water and nutrients absorption. Given that the nutrient solution was neither replaced nor refilled with water, in order to achieve the highest possible nutrient- and water-use efficiency, probably the increase of EC during the last week of the experiment is attributable to a higher rate of water absorption by the roots than of the dissolved nutrients and consequently resulted in a more concentrated growing solution. The form of nitrogen exerts the main impact on the pH levels around the rhizosphere [29]. When supplied in nitrate form, roots release anions that increase the pH, while when it is provided in ammonium form, roots release cations that decrease the pH of the rhizosphere [30]. However, when nitrogen is supplied in the form of amino acids, the pH may or may not change, depending on the amino acid type and charge [31]. As long as the nitrate form of nitrogen was initially high in the NS and rapidly absorbed, as well, the pH increased because the absorption of anions exceeded that of cations in meq basis [32].

### 3.2. Plant Growth

Plant growth was determined by fresh weight, plant height, dry matter and length of roots (Table 1). Neither fresh weight nor dry matter of both mint species was affected by the addition of the PHs solution; however, the plant height of peppermint decreased at both PHs rates, and of spearmint only at 0.5%, rendering the shoots shorter. Root length decreased linearly ($r = 0.959$, $p < 0.001$) to the PHs solution concentration, and, indeed, at the high dosage, the root growth was almost halted. The suppressed height and root growth may indicate ammonium toxicity [33] induced by the high concentration of total nitrogen of the commercial formulation used and the conversion of the amino

acids to ammonium nitrogen. Indeed, according to the manufacturer [34], the commercial PH solution contains 12.5% ($w/w$) amino acids that correspond to 1.7% total nitrogen. This consequently means that 0.25 and 0.50% of Amino16 solution in the 50 L tank may even correspond up to 3 and 6 mM $NH_4$-N, which is almost a four to eight times higher concentration in the nutrient solution than of the control treatment. Others [21] associated this effect to an auxin-like activity of the PHs, while observing a decrease in root length of watercress after the application of two PH fertilizers. Colla et al. [18] reported that the PHs exhibit a bio-stimulant functional role, increasing crop yield, nutrient uptake and water absorption and subsequently promote plant growth. However, in the current study, plant growth was not affected either positively or adversely by the addition of PHs in the nutrient solution. This may be attributed to either the abnormal root growth or to the low rate of PHs applied and needs to be further investigated, as there are contradictory results in the published literature. In particular, studies on onion [35], spearmint [14], basil [36] and leafy radish [37] reported a positive effect of amino acids on plant growth, while other ones on pak-choi [38] and rocket [22] revealed a negative or no effect. Apparently, different species exhibit different physiological responses to the application of bio-stimulants, which are, in turn, affected by the application mode and the dosage used [18].

**Table 1.** Means of plant growth and nutritional composition characteristics of peppermint and spearmint plants grown in floating system in an NS in which 0, 0.25 or 0.50% of a commercial PH solution (Amino16®) was added.

| | Peppermint | | | | | | Spearmint | | | | | |
|---|---|---|---|---|---|---|---|---|---|---|---|---|
| | Amino Acids Concentration | | | | | | | Amino Acids Concentration | | | | |
| | 0% | | 0.25% | | 0.50% | | *p* | 0% | | 0.25% | | 0.50% | | *p* |
| Fresh weight (g) | 10.2 [a] | | 8.52 | | 7.53 | | ns [b] | 8.05 | | 8.37 | | 6.23 | | ns |
| Plant height (cm) | 22.83 | a [c] | 19.46 | b | 18.97 | b | * | 24.06 | a | 24.82 | a | 21.21 | b | * |
| Root length (cm) | 12.97 | a | 4.48 | b | 0.74 | c | *** | 11.48 | a | 3.98 | b | 0.94 | c | *** |
| Dry matter (%) | 14.2 | | 14.7 | | 15.5 | | ns | 17.2 | | 18.8 | | 19.3 | | ns |
| Total soluble solids (%) | 9.5 | | 10.0 | | 10.5 | | ns | 9.5 | | 10.7 | | 10.3 | | ns |
| Nitrates (mg/kg f.w.) | 828.5 | a | 647.4 | a | 298.7 | b | *** | 797.3 | a | 525.8 | b | 179.8 | c | *** |
| Total antioxidant capacity (mg AEAC/100 g f.w.) | 110.3 | | 95.6 | | 134.6 | | ns | 65.0 | b | 71.3 | b | 172.4 | a | ** |
| Total soluble phenols (mg GAE/kg f.w.) | 0.467 | | 0.340 | | 0.560 | | ns | 0.367 | b | 0.407 | b | 0.760 | a | * |
| Total chlorophyll (μg/g f.w.) | 1138.4 | b | 1327.0 | a | 1248.5 | ab | * | 1173.4 | b | 1379.8 | a | 1316.7 | a | * |
| Total carotenoids (μg/g f.w.) | 200.4 | | 231.4 | | 225.6 | | ns | 204.7 | b | 247.9 | a | 246.5 | a | * |
| Essential oil (ml/100 g f.w.) | 0.34 | b | 0.35 | b | 0.43 | a | * | 0.19 | b | 0.20 | b | 0.24 | a | * |

[a] Values are means of three replicates. [b] *, **, *** shows significant differences at 0.05, 0.01 and 0.001 probability levels; ns shows non-significant differences. [c] Values in the same row followed by different letters differ significantly by LSD test ($p < 0.05$).

### 3.3. Nutritional Composition

Apart from the plant physiology responses, the PHs may be also used to improve the product quality, by inducing the primary and secondary metabolism in plants, resulting in accumulation of valuable phytochemicals in plant tissues. The nutritional composition of peppermint and spearmint shoots was assessed by determining the total soluble solids, nitrate, total soluble phenols, total chlorophyll, total carotenoids and essential oil content, as well as the total antioxidant capacity (Table 1). In both species, total soluble solids content was not affected by the addition of the PHs solution, implying that no osmotic stress was induced in the plants.

Nitrate content decreased in peppermint plants grown with the high (0.5%) PH concentration and in spearmint plants grown with both PH-enriched NSs. This decrease may be attributed to the consistent and continuous supply of ammonium nitrogen [39], or to the accumulation of amino acids in the plant tissues that regulate the uptake and reduction of nitrate [37]. It has been suggested that the presence of more than one nitrogen form around the rhizosphere adversely affects the absorption of each other; namely, ammonium ions prevent the absorption of nitrate ions, and, in turn, the external application of amino acids (PHs) inhibits the absorption of both ammonium and nitrate ions [19]. The observed

decrease in nitrate content is in accordance with the results of Wang et al. [37] on pak-choi, Mobini et al. [35] on onion bulbs, Vernieri et al. [22] on rocket and Tsouvaltzis et al. [40,41] on lettuce. The accumulation of nitrates in leafy vegetables is a phenomenon that has drawn the attention of the medical society, since the high nitrate content is associated with severe human-health risks. Thus, the use of practices towards the reduction of nitrate accumulation is of high importance, especially when it is simultaneously maintaining a high crop yield and enhanced product quality. Despite the low consumption of fresh aromatic herbs, most of these green leafy products accumulate high nitrate content in their tissues [42]. That high nitrate content in fresh tissue results is further increased nitrate concentration after drying and during storage [43]. Thus, given that they are mostly consumed as dry tissue, the implementation of cultivation practices that prevent nitrate accumulation would be widely acknowledged.

Aromatic plants contain a plethora of antioxidants, with the most important being phenolic and volatile compounds, as well as others, such as ascorbic acid, carotenoids and chlorophylls, which are of minor importance [2,4,44]. While chlorophylls may not be exhibit strong antioxidant activity, their high concentration in plant tissues contributes to the overall plant antioxidant capacity [45]. Total antioxidant capacity and total soluble phenol content in peppermint were not affected by either rate of PHs solution; however, in spearmint, there was a significant increase of both components when the high dosage of PHs solution was applied in the NS (Table 1). The increase of total antioxidant capacity and total soluble phenols content may be attributed to either the bio-stimulant properties of the amino acids or to the ammonium stress. It is not likely that the overall increased nitrogen nutrition that peppermint and spearmint received is the reason for that increase, as long as other researchers have observed a negative or lack of effect of high nitrogen supply on the antioxidant capacity and total phenol content in basil and spearmint plants [16,46]. Sharafzadeh [47] reported an increase in total phenol content of thyme plants after application of 50–75 mg/kg of nitrogen; however, it later returned to the initial levels, while further increasing the nitrogen levels in the soil. It is possible that lower nitrogen rates during fertilization were not enough to be used in the secondary metabolism. Nutrient toxicity is one of several stress factors that induce the secondary metabolism [48–50], and therefore it is possible that this increase was a consequence of ammonium toxicity.

Total chlorophyll significantly increased in peppermint when the low dose of PHs solution was added, but total carotenoids were not affected by either dose. However, in spearmint plants, both total chlorophyll and total carotenoids increased significantly when either dose of PHs solution was added in the NSs. High chlorophyll content improves leaf color, and since color is one of the most important characteristics of fresh fruit and vegetable quality, a greener product will be more attractive to the customers in the retail market [51]. The increased chlorophyll content may be attributed to the increased nitrogen concentration in the NS, as it is an integral constituent of the chlorophyll molecule [29]. However, *Arabidopsis thaliana* plants subjected to mild ammonium toxicity showed an accumulation of chlorophyll in their tissues [52], indicating that there was indeed an ammonium toxicity due to the conversion of the amino acids to ammonium nitrogen. Increased nitrogen fertilization has been shown to increase total chlorophyll in spearmint [16], and chlorophylls a and b and carotenoids in parsley [53].

While most phytochemicals in peppermint were not affected by the PHs supplement, the essential oil content increased significantly by 26.5%, when the high dose of PHs solution was used. In spearmint, a similar increase was observed (26.3%) (Table 1). A higher essential oil content was also observed by Reference [14], in spearmint (by 25%), when the plants were grown in deep flow technique in NS containing a mixture of amino acids, but not in Japanese mint. This increase was even higher when the spearmint was grown in soil. However, when single amino acids were used instead of mixtures, adverse effects were induced, such as a decrease in the essential oil content that was observed in sweet basil when glycine or glutamine was used as replacements of ammonium or nitrate [35]. Mineral nutrition is an important factor in regard to essential oil content [54–56]. Increasing nitrogen

fertilization raised the essential oil content in thyme, basil and peppermint [46,56,57]; however, in some cases, negative or no effects at all were reported in basil, oregano, sage and spearmint [16,54,55,58]. However, most researchers have stated that the increase of essential oil content is a result of the increased biomass production and photosynthetic rate [56], both of which are positively affected by the increased nitrogen fertilization. In the present study, no such increase of fresh or dry biomass was observed, confirming the speculation that amino acids not only act as a nitrogen source but as bio-stimulants, as well.

### 3.4. Essential Oil Composition

In total, thirty-three constituents were identified in peppermint, accounting for 99.78–99.82% of the total extracted oil; and thirty-nine were identified in spearmint, accounting for 99.24–99.52% of the total oil (Table 2). The most abundant detected compounds were oxygenated monoterpenes in peppermint and monoterpene hydrocarbons in spearmint. The main compounds in peppermint were menthone (55.44–55.84%), menthol (14.68–15.16%), menthofuran (5.43–7.11%), isomenthone (6.01–6.76%), 1,8-cineole (3.95–4.26%), limonene (2.22–2.36%), pulegone (1.70–1.86%), terpinen-4-ol (1.09–1.27%) and germacrene-D (1.01–1.09%). The main compounds in spearmint were carvone (39.87–41.86%), limonene (16.48–17.57%), 1,8-cineole (16.44–17.48), myrcene (4.61–4.65%), germacrene-D (3.86–4.35%), β-caryophyllene (3.65–3.92%), β-pinene (2.49–2.65%), sabinene (1.66–1.77%), α-pinene (1.22–1.27%) and *trans*-carveol (0.76–1.16%). None of the major components in both peppermint and spearmint were affected by the addition of PHs solution in the nutrient solution. The only effect observed was in the contents of 3-octanol, terpinolene, caryophyllene oxide and 13-epi-manool oxide in spearmint, but only 3-octanol levels were higher than 0.1%, to address a significant effect of the PHs. The relative concentration of these compounds is so exceptionally low that they do not have any impact on the aroma or flavor. The addition of amino acids or nitrogen in the NS of hydroponically grown herbs may affect positively or negatively the chemical composition of aromatic plants, but such changes may be species-specific and in relation to the amino acid or nitrogen concentration used, as observed in spearmint and Japanese mint [14,16]. In the present study, the low concentration of PHs solution used may not have been enough to induce the secondary metabolism to such an extent that it would alter the overall composition of the essential oil.

**Table 2.** Essential oil composition of peppermint and spearmint plants grown in floating system in which 0, 0.25 or 0.50% of a commercial PH solution (Amino16®) was added.

| No. | Compounds | RI [a] | Peppermint | | | | Spearmint | | | | | | |
| | | | Amino Acids Concentration | | | | Amino Acids Concentration | | | | | | |
| | | | 0% | 0.25% | 0.50% | *p* | 0% | | 0.25% | | 0.50% | | *p* |
|---|---|---|---|---|---|---|---|---|---|---|---|---|---|
| 1 | α-thujene | 929 | 0.03 [b] | 0.03 | 0.03 | ns [c] | 0.02 | | 0.01 | | 0.02 | | ns |
| 2 | α-pinene | 935 | 0.60 | 0.64 | 0.60 | ns | 1.23 | | 1.22 | | 1.27 | | ns |
| 3 | camphene | 950 | - | - | - | | 0.02 | | 0.01 | | 0.01 | | ns |
| 4 | sabinene | 974 | 0.42 | 0.44 | 0.43 | ns | 1.66 | | 1.69 | | 1.77 | | ns |
| 5 | β-pinene | 977 | 0.90 | 0.92 | 0.90 | ns | 2.52 | | 2.49 | | 2.65 | | ns |
| 6 | 1-octen-3-ol | 982 | 0.02 | 0.01 | 0.02 | ns | - | | - | | - | | |
| 7 | myrcene | 992 | 0.25 | 0.24 | 0.24 | ns | 4.65 | | 4.65 | | 4.61 | | ns |
| 8 | 3-octanol | 996 | 0.07 | 0.05 | 0.07 | ns | 0.72 | b [d] | 0.76 | b | 0.86 | a | ** |
| 9 | pseudolimonene | 1004 | - | - | - | | 0.11 | | 0.09 | | 0.11 | | ns |
| 10 | α-terpinene | 1016 | 0.26 | 0.22 | 0.23 | ns | 0.04 | | 0.02 | | 0.02 | | ns |
| 11 | *o*-cymene | 1025 | 0.03 | 0.03 | 0.04 | ns | - | | - | | - | | |
| 12 | limonene | 1029 | 2.22 | 2.29 | 2.36 | ns | 17.57 | | 16.48 | | 16.80 | | ns |
| 13 | 1,8-cineole | 1032 | 4.14 | 3.95 | 4.26 | ns | 16.44 | | 17.48 | | 17.41 | | ns |
| 14 | *cis*-β-ocimene | 1040 | 0.14 | 0.12 | 0.12 | ns | 0.61 | | 0.62 | | 0.64 | | ns |
| 15 | *trans*-β-ocimene | 1051 | - | - | - | | 0.12 | | 0.13 | | 0.14 | | ns |
| 16 | γ-terpinene | 1060 | 0.46 | 0.39 | 0.42 | ns | 0.07 | | 0.04 | | 0.05 | | ns |
| 17 | cis-sabinene hydrate | 1068 | 0.37 | 0.60 | 0.51 | ns | 0.01 | | 0.03 | | 0.03 | | ns |
| 18 | terpinolene | 1087 | 0.12 | 0.10 | 0.11 | ns | 0.09 | a | 0.08 | b | 0.08 | b | * |

**Table 2.** *Cont.*

| No. | Compounds | RI [a] | Peppermint Amino Acids Concentration 0% | 0.25% | 0.50% | p | Spearmint Amino Acids Concentration 0% | | 0.25% | | 0.50% | | p |
|---|---|---|---|---|---|---|---|---|---|---|---|---|---|
| | Monoterpene hydrocarbons (%) | | 10.00 | 10.04 | 10.33 | | 45.88 | | 45.79 | | 46.47 | | |
| 19 | linalool | 1100 | 0.10 | 0.10 | 0.11 | ns | 0.11 | | 0.10 | | 0.11 | | ns |
| 20 | *trans*-pinocarveol | 1139 | - | - | - | | 0.15 | | 0.14 | | 0.17 | | ns |
| 21 | menthone | 1158 | 55.44 | 55.50 | 55.84 | ns | - | | - | | - | | |
| 22 | menthofuran | 1164 | 6.49 | 7.11 | 5.43 | ns | - | | - | | - | | |
| 23 | isomenthone | 1166 | 6.25 | 6.01 | 6.76 | ns | - | | - | | - | | |
| 24 | neomenthol | 1167 | 0.12 | 0.11 | 0.12 | ns | - | | - | | - | | |
| 25 | δ-terpineol | 1167 | - | - | - | | 0.45 | | 0.45 | | 0.52 | | ns |
| 26 | menthol | 1176 | 15.16 | 14.68 | 14.79 | ns | - | | - | | - | | |
| 27 | terpinen-4-ol | 1177 | 1.27 | 1.09 | 1.20 | ns | 0.13 | | 0.10 | | 0.13 | | ns |
| 28 | neoisomenthol | 1182 | 0.07 | 0.06 | 0.07 | ns | - | | - | | - | | |
| 29 | α-terpineol | 1189 | 0.14 | 0.11 | 0.14 | ns | 0.91 | | 0.86 | | 0.94 | | ns |
| 30 | dihydrocarveol | 1193 | - | - | - | | 0.11 | | 0.13 | | 0.15 | | ns |
| 31 | *trans*-dihydrocarvone | 1194 | - | - | - | | 0.12 | | 0.10 | | 0.16 | | ns |
| 32 | *trans*-carveol | 1219 | - | - | - | | 1.08 | | 0.76 | | 1.16 | | ns |
| 33 | pulegone | 1238 | 1.76 | 1.80 | 1.70 | ns | - | | - | | - | | |
| 34 | carvone | 1247 | - | - | - | | 40.64 | | 41.86 | | 39.87 | | ns |
| 35 | piperitone | 1253 | 0.49 | 0.51 | 0.54 | ns | - | | - | | - | | |
| 36 | dihydroedulan II | 1284 | - | - | - | | 0.13 | | 0.12 | | 0.16 | | ns |
| 37 | menthyl acetate | 1294 | 0.42 | 0.55 | 0.55 | ns | - | | - | | - | | |
| | Oxygenated monoterpenes (%) | | 87.71 | 87.64 | 87.25 | | 43.83 | | 44.61 | | 43.37 | | |
| 38 | α-copaene | 1376 | - | - | - | | 0.02 | | 0.02 | | 0.02 | | ns |
| 39 | β-bourbonene | 1383 | - | - | - | | 0.49 | | 0.47 | | 0.52 | | ns |
| 40 | β-elemene | 1391 | - | - | - | | 0.18 | | 0.17 | | 0.18 | | ns |
| 41 | β-caryophyllene | 1415 | 0.60 | 0.63 | 0.67 | ns | 3.82 | | 3.65 | | 3.92 | | ns |
| 42 | α-humulene | 1451 | - | - | - | | 0.17 | | 0.16 | | 0.18 | | ns |
| 43 | *cis*-β-farnesene | 1461 | 0.11 | 0.11 | 0.12 | ns | 0.11 | | 0.09 | | 0.10 | | ns |
| 44 | germacrene-D | 1479 | 1.01 | 1.03 | 1.09 | ns | 4.35 | | 4.02 | | 3.86 | | ns |
| 45 | bicyclogermacrene | 1492 | 0.09 | 0.08 | 0.09 | ns | 0.37 | | 0.33 | | 0.31 | | ns |
| 46 | δ-cadinene | 1525 | - | - | - | | 0.09 | | 0.08 | | 0.08 | | ns |
| | Sesquiterpene hydrocarbons (%) | | 1.82 | 1.86 | 1.97 | | 9.61 | | 8.99 | | 9.16 | | |
| 47 | caryophyllene oxide | 1584 | - | - | - | | 0.04 | a | 0.01 | b | 0.03 | ab | * |
| 48 | viridiflorol | 1591 | 0.26 | 0.24 | 0.26 | ns | - | | - | | | | |
| | Oxygenated sesquiterpenes (%) | | 0.26 | 0.24 | 0.26 | | 0.04 | | 0.01 | | 0.03 | | |
| 49 | 13-epi-manool oxide | 2012 | - | - | - | | 0.12 | b | 0.12 | b | 0.20 | a | * |
| | Oxygenated diterpenes (%) | | 0.00 | 0.00 | 0.00 | | 0.12 | | 0.12 | | 0.20 | | |
| | Total (%) | | 99.79 | 99.78 | 99.81 | | 99.48 | | 99.52 | | 99.24 | | |

[a] RI: retention indices relative to n-alkanes on a DB-5 column. [b] Values are means of three replicates. [c] *, ** shows significant differences at 0.05, 0.01 and 0.001 probability levels; ns shows non-significant differences. [d] Values in the same row followed by different letters differ significantly by LSD test ($p < 0.05$).

### 3.5. Polyphenolic Composition

Twenty-six polyphenolic compounds were identified as belonging to six groups: benzoic acid derivatives, coumarins, flavones, flavanones, flavonols and phenylpropanoids. The main polyphenolic compounds were rosmarinic acid (253–402 mg/100 g f.w. in peppermint and 400–437 mg/100 g f.w. in spearmint), hesperidin (129–147 mg/100 g f.w. and 87–105 mg/100 g f.w.), quercetin-4-O-glucoside (40–45 mg/100 g f.w. and 26–31 mg/100 g f.w.), caftaric acid (19–29 mg/100 g f.w. and 13–16 mg/100 g f.w.), cryptochlorogenic acid (16–21 mg/100 g f.w. and 15–20 mg/100 g f.w.) and caffeic acid (10–12 mg/100 g f.w. and 13–14 mg/100 g f.w.) (Table 3). The addition of PHs solution in the NS did not affect most of the phenolic compounds, except for *t*-coutaric acid in peppermint, having its concentration increased, and the 2,6-dihydroxybenzoic acid and arbutin in spearmint, with their concentration decreased. Areias et al. [59] suggested that, as long as peppermint does not have a polyphenolic marker compound, it would be best to be characterized by

the highest possible number of compounds that can be identified, with rosmarinic acid, hesperidin, eriocitrin, eriodictyol-7-*O*-rutinoside and luteolin-7-*O*-rutinoside as the main ones. The qualitative composition of spearmint methanolic extracts demonstrated in the present study is consistent with previous reports [60].

**Table 3.** Polyphenolic composition of peppermint and spearmint plants grown in floating system in which 0, 0.25 or 0.50% of a commercial PH solution (Amino16®) was added.

| No. | Compounds (mg/100 g d.w.) | Peppermint Amino Acids Concentration | | | | | | | | | Spearmint Amino Acids Concentration | | | | | | | | |
|---|---|---|---|---|---|---|---|---|---|---|---|---|---|---|---|---|---|---|---|
| | | 0% | | 0.25% | | 0.50% | | *p* | | 0% | | 0.25% | | 0.50% | | *p* | | | |
| 1 | Vanillic acid | 0.34 [a] | | 0.46 | | 0.38 | | ns [b] | | 0.84 | | 0.69 | | 0.82 | | ns | | | |
| 2 | 2,6-dihydroxybenzoic acid | 0.68 | | 0.32 | | 0.40 | | ns | | 0.77 | a [c] | 0.49 | b | 0.82 | a | * | | | |
| 3 | Syringaldehyde | 0.03 | | 0.04 | | 0.03 | | ns | | 0.06 | | 0.06 | | 0.05 | | ns | | | |
| 4 | Daphetin | 0.13 | | 0.11 | | 0.10 | | ns | | 0.13 | | 0.13 | | 0.17 | | ns | | | |
| 5 | Caffeic acid | 11.79 | | 10.28 | | 10.41 | | ns | | 13.23 | | 13.81 | | 14.25 | | ns | | | |
| 6 | Ferulic acid | 0.23 | | 0.26 | | 0.25 | | ns | | 0.29 | | 0.27 | | 0.30 | | ns | | | |
| 7 | Caftaric acid | 19.35 | | 29.05 | | 25.14 | | ns | | 13.73 | | 12.58 | | 16.45 | | ns | | | |
| 8 | Neochlorogenic acid | 5.24 | | 7.57 | | 7.41 | | ns | | 3.06 | | 3.31 | | 3.50 | | ns | | | |
| 9 | Chlorogenic acid | 4.62 | | 5.89 | | 4.89 | | ns | | 4.47 | | 4.44 | | 5.23 | | ns | | | |
| 10 | Rosmarinic acid | 287.81 | | 402.33 | | 253.35 | | ns | | 417.62 | | 400.47 | | 437.49 | | ns | | | |
| 11 | Sinapyl alcohol | 0.68 | | 0.56 | | 0.75 | | ns | | 0.56 | | 0.57 | | 0.58 | | ns | | | |
| 12 | Fertaric acid | 1.80 | | 1.87 | | 1.82 | | ns | | 2.60 | | 2.72 | | 2.23 | | ns | | | |
| 13 | *t*-coutaric | 0.76 | b | 1.77 | a | 1.54 | a | * | | 1.14 | | 1.04 | | 1.05 | | ns | | | |
| 14 | Apigenin | 1.57 | | 1.31 | | 1.34 | | ns | | 0.55 | | 0.98 | | 0.81 | | ns | | | |
| 15 | Luteolin | 7.50 | | 6.25 | | 5.77 | | ns | | 1.85 | | 2.61 | | 2.72 | | ns | | | |
| 16 | Luteolin-7-*O*-glucoside | 0.88 | | 0.92 | | 0.81 | | ns | | 1.18 | | 1.63 | | 1.43 | | ns | | | |
| 17 | Hesperidin | 128.98 | | 146.85 | | 134.46 | | ns | | 88.72 | | 86.73 | | 104.79 | | ns | | | |
| 18 | Apigenin-7-glucoside | 0.21 | | 0.22 | | 0.13 | | ns | | 0.43 | | 0.66 | | 0.55 | | ns | | | |
| 19 | Naringenin | 1.90 | | 1.93 | | 2.00 | | ns | | 1.08 | | 1.30 | | 1.48 | | ns | | | |
| 20 | Kaempferol-3-glucoside | 0.02 | | 0.01 | | 0.02 | | ns | | 0.02 | | 0.03 | | 0.03 | | ns | | | |
| 21 | Arbutin | 0.07 | | 0.10 | | 0.10 | | ns | | 0.13 | a | 0.08 | b | 0.08 | b | * | | | |
| 22 | Syringic acid | 0.76 | | 0.49 | | 0.49 | | ns | | 0.47 | | 0.63 | | 0.67 | | ns | | | |
| 23 | Cryptochlorogenic acid | 16.22 | | 21.03 | | 16.94 | | ns | | 15.40 | | 16.69 | | 19.78 | | ns | | | |
| 24 | Quercetin-3-glucoside | 0.23 | | 0.30 | | 0.29 | | ns | | 0.34 | | 0.25 | | 0.39 | | ns | | | |
| 25 | Quercetin-4-*O*-glucoside | 40.03 | | 45.07 | | 40.84 | | ns | | 26.43 | | 25.90 | | 31.15 | | ns | | | |
| 26 | Rutin | 1.09 | | 1.31 | | 1.24 | | ns | | 2.47 | | 1.09 | | 2.15 | | ns | | | |

[a] Values are means of three replicates. [b] * shows significant differences at 0.05, 0.01 and 0.001 probability levels; ns shows non-significant differences. [c] Values in the same row followed by different letters differ significantly by LSD test (*p* < 0.05).

The changes on the accumulation of the various phenolic compounds under the amino acid treatments observed in our study could be explained with the Carbon–Nutrient Balance (CNB) hypothesis [61]. According to this hypothesis, increased nutrient uptake reduces the C/N ratio within the plant; C-based secondary metabolism declines as growth receives allocation priority. Two major parameters that could affect the C/N balance within the plants and, consequently, secondary metabolism are the light intensity and vegetative stage (vegetative to reproductive structures). Increased light intensity is predicted to increase net photosynthesis, thereby increasing the C/N ratio within the plant, and concentrations of C-based secondary metabolites. In addition, while reproductive structures contribute directly and indirectly to their own carbon requirements, the substantial nutrient investment required for their maturation is translocated from other parts of the plant. As a result, the diversion of nutrients from vegetative to reproductive structures may be proportionally greater than that of carbon. According to all the above, the increased content of specific phenolic metabolites at the stage before flowering and/or under the high amino acid treatment could be attributed to the differential partitioning of nutrients to reproductive structures resulting in an increase of the C/N ratio in the foliage and the relatively high light density during the experimental period (spring/summer).



## 4. Conclusions

In conclusion, in the present study, the addition of a PH solution in the NS of peppermint and spearmint hydroponic floating production system reduced plant height, without affecting the fresh weight; enhanced nutritional composition by reducing the nitrate content and increasing total chlorophyll and antioxidant components, such as total soluble phenols and carotenoids; and increased the essential oil content. Simultaneously, neither the essential oil nor the polyphenolic composition profile was altered. According to the above results, the PH solution is advisable to be used during peppermint and spearmint production in a soilless floating system, in order to improve the product quality, without inducing adverse effects on fresh or dry weight and on the essential oil and phenolic profile.

**Author Contributions:** Conceptualization, D.-C.A., P.T. and A.S.S.; methodology, P.T., D.-C.A., D.S.K. and E.S.; software, P.T., D.-C.A. and E.S.; formal analysis, A.S.S., P.C. and S.M.; investigation, D.S.K. and D.-C.A.; data curation, P.T., D.-C.A. and E.S.; writing—original draft preparation, P.T., D.S.K., D.-C.A. and E.S.; writing—review and editing, S.M., P.C. and A.S.S.; supervision, P.T. and A.S.S. All authors have read and agreed to the published version of the manuscript.

**Funding:** This research received no external funding.

**Data Availability Statement:** Data available on request due to privacy restrictions.

**Conflicts of Interest:** The authors declare no conflict of interest.

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
