# Peer review of "Protein Hydrolysates Supplement in the Nutrient Solution of Soilless Grown Fresh Peppermint and Spearmint as a Tool for Improving Product Quality"

_agronomy, doi:10.3390/agronomy11020317_

Round 1
Reviewer 1 Report
General comments
The manuscript deals with the effects of biostimulants, namely the commercially available product of Amino16®, on plants of Mentha species grown in a hydroponic system. Plants were evaluated with respect to biometric values and contents of secondary metabolites such as antioxidants, essential oils and polyphenols.
The manuscript addresses an interesting subject, as many biostimulant products are currently in the market, often with limited evidence of their effects. The manuscript is generally well-written and correct when it comes to format. The methodologies used are sound. The discussion is elaborate and the results are discussed in relation to relevant references. The conclusions drawn from the results are in general reasonable, with some exemptions (see detailed comments). The reference list seem correct and complete.
Detailed comments
Line 28: Please arrange key words alphabetically
- 31: Please italicize the scientific names. Throughout paper.
- 93: How was the temperature and humidity measured? Please state the equipment used and the frequency of measurements.
- 93: Please use ° for degrees (throughout paper)
- 95: Replace “N-NO3” and N-NH4” with “NO3-N” and “NH4-N”. Note the use of subscript.
- 97: Which concentration was recommended by the manufacturer?
- 116: Remove redundant space between “70” and “°C” Also in other places in the paper.
- 169-170: Please be a bit more elaborate on the statistics methods used.
- 177-178: What is your explanation for this increase in EC?
- 209-210: I am a bit doubtful about this statement. You have in your solution from the beginning only 14 mg/L NH4-N, which is far below potentially toxic levels. Do you really supply so much biostimulants that NH4 levels could reach critical levels? Please do some calculations and perhaps insert additional references.
- 381-383: Are floating raft production common in commercial production?
L- 381-383: I do not feel that this is a sound conclusion, with respect to the results, as there were not really that great impact by treatment on the secondary metabolites.
Table 1: As you are referring in the text to plant “compactness”, I would suggest that you insert some measurement of plant compactness in this table, for example relative weight in g/cm. If no differences between the treatments for this figure, I think one cannot say that the plants in a specific treatment were “more compact”.
Table 1: I would be a bit cautious with stating that there “were no differences in fresh weight”. Indeed, the statistic methods used (which? Please clarify) indicates no differences, but I would say mainly because of few replicates. There is a clear trend, with differences up to 26%, in fresh weight, so I would say that it is quite obvious that the biostimulant actually inhibited growth.
- 377: Se my opinion in my comments for table 1. Please clarify that there was a trend of decreased FW for plants treated with the product.
Author Response
Our point to point response to the reviewer's comments are included in the attached file.

Reviewer 2 Report
Manuscript presented for review with title: “Protein hydrolysates supplement in the nutrient solution of soilless grown fresh peppermint and spearmint as a tool for improving product quality” is really interesting and very important. The experiment was planned very carefully. I’m impress of excellent work made by authors. The Introduction section includes all necessary information about examined objects and problems.
The collected experimental material and used methods do not raise any objections.
I have some comments to authors:
page 1, line 14-15: all plants Latin' names should be written by italic type. Please change it here and in every place in whole manuscript, where those names are used.
page 3, lines: 114-118, why temperature of dry mater evaluation was only 72oC? According to my knowledge to obtain total dry matter it should be use 105 oC. Of course is not possible to use that material for another analysis.
page 5, line 205: “…the addition of the PHs solution..: it should not be pHs? If yes, please correct it in whole sub-section and manuscript text.
Table 3: how many replication was used for mean value calculation (n=?) if in table is mean value why SD or SE are not presented? Please add it to each mean value.
The discussion section presents a very good comparison of the obtained results with other results available in the data basis.
The obtained conclusions are clear but their presentation is too long.
General opinion: I think, that presented manuscript is a very valuable and should be ok after little correction according my remarks in Agronomy.
Author Response

(The authors gave the same response as above.)
